# Influence of Different Antiseizure Medications on Vascular Risk Factors in Children with Epilepsy

**DOI:** 10.3390/children9101499

**Published:** 2022-09-30

**Authors:** Doaa M. Mahrous, Asmaa N. Moustafa, Mahmoud M. Higazi, Aliaa M. Higazi, Reem A. AbdelAziz

**Affiliations:** 1Department of Pediatrics, Faculty of Medicine, Minia University, Minia 61519, Egypt; 2Department of Radiology, Faculty of Medicine, Minia University, Minia 61519, Egypt; 3Department of Clinical and Chemical Pathology, Faculty of Medicine, Minia University, Minia 61519, Egypt

**Keywords:** epilepsy, antiseizure medications, carotid intima-media thickness, carotid stiffness, homocysteine

## Abstract

Many studies have proposed that plasma homocysteine levels are increased as a side effect with the prolonged use of antiseizure medications. This is associated with an increase in carotid intima media thickness; hence, it increases the threat of atherosclerosis at a young age. We aimed to assess serum levels of homocysteine in epileptic children on long-standing antiseizure medications and its association with increased occurrence of cardiovascular disease. The study included 60 epileptic children aged between 2 and 15 years old who visited our pediatric neurology outpatient clinic and 25 apparently healthy children served as a control group. All included children were subjected to careful history taking, clinical examination, anthropometric measures, laboratory investigations including serum homocysteine levels and lipid profile, along with radiological assessment involving carotid intima media thickness and carotid stiffness. Results demonstrated a significant increase in the serum levels of homocysteine, carotid intima media thickness, and carotid stiffness in children on monotherapy of old generation antiseizure medications and polytherapy than that in children on monotherapy of new generation antiseizure medications and control children. Epileptic children on old generation and polytherapy antiseizure medications have an increased risk for cardiovascular diseases and need follow up for early intervention when needed.

## 1. Introduction

Epilepsy is a long-term neurological disorder marked by recurring seizures caused by excessive neuronal discharges [1]. Antiseizure medications (ASMs) are divided into first, second, and third generations. The optimal medication for each individual should be chosen based on various considerations, including cost, range of activity, the possibility of dose-related and major adverse effects, drug interactions, and other factors [2].

Homocysteine (Hcy) is sulfhydryl-containing amino acid. It is a byproduct in the normal metabolism of the amino acid methionine to cysteine [3]. Many studies have demonstrated that the prolonged use of certain ASMs raises homocysteine levels, which is related to vascular endothelial damage and, therefore, raises the chances of atherosclerosis and cardiovascular diseases [4]. Elevated serum homocysteine levels increase oxidative stress, thus, cause vascular inflammation, increase platelet aggregation, induce thrombosis and coagulation disorders, and consequently, produce endothelial damage. Endothelial dysfunction is thought to be the primary mechanism through which homocysteine induces vascular diseases. Moreover, endothelial dysfunction relates to the severity of cardiovascular disorders due to the reduced bioavailability of nitric oxide because of higher asymmetric dimethylarginine (ADMA) concentrations [5].

Arterial stiffness is one of the earliest manifestations of the functional and structural changes in the wall of the carotid artery. Therefore, carotid intima-media thickness (CIMT) is an approved early marker for subclinical carotid atherosclerosis and for cardiovascular disease prediction. Hence, pharmacological therapy should be started early [6].

The purpose of our work is to evaluate the serum levels of homocysteine and lipid profiles in epileptic children who are on long-standing ASMs and its correlation with increased incidence of cardiovascular disease by measuring carotid intima-media thickness (CIMT) and carotid stiffness.

## 2. Materials and Methods

### 2.1. Study Sample

This is a case-control study. Using the Epi Info program, and at a 95% confidence level, 51 subjects were required to establish the study. We recruited 60 epileptic children who had regular follow up in the Pediatric Neurology Outpatient Clinic, Minia University Children and Maternity Hospital, Egypt, in addition, 25 apparently healthy children matching age and sex with the diseased group were included as a control group. They were grouped as follows:

Group A: 20 epileptic children; received a single old antiseizure medication (valproic acid, carbamazepine, and phenytoin) for at least one year. Group B: 20 epileptic children; received a single new antiseizure medication (levetiracetam, oxcarbazepine, lamotrigine, and topiramate) for at least one year. Group C: 20 epileptic children; received two or more ASMs whatever its generation (with at least one agent related to old ASMs) for at least one year. Group D: 25 apparently healthy children; matching age and sex with patient groups. 

Patients with ages ranging from 2 to 15 years and those with idiopathic epilepsy on regular ASMs for at least one year were included in the study. Any patients with secondary epilepsy, on ASMs for less than one year, with any vascular disease that may affect elasticity or thickness of vessels such as hypertension, with any metabolic disease that may affect lipid profile such as obesity, primary biliary cirrhosis, primary sclerosing cholangitis, and type I hyperlipoproteinemia, with any disease that may cause hyperhomocysteinemia such as leukemia, psoriasis, sickle cell anemia, polycythemia Vera, and idiopathic thrombocytosis were excluded from our study.

All included children underwent careful history taking, including name, age, sex, type of convulsion (focal or generalized), type of treatment (old, new ASMs, or polytherapy), duration of treatment, family history, response to treatment, last fits, and the number of status epilepticus. Clinical examinations, including vital data, heart rate, blood pressure, and anthropometric measures, including body weight, height, and BMI, were conducted on all our patients. The Global Assessment of Severity of Epilepsy (GASE) Scale was also carried out for each epileptic child [7]. 

### 2.2. Laboratory Investigations

Five milliliters of venous blood were withdrawn into appropriately labeled tubes from each patient and control subject under complete aseptic conditions. The blood samples were left to clot for 30 min at room temperature, then samples were centrifuged at 3000 rpm for 15 min for serum separation. Serum samples were collected in separate Eppendorf tubes and stored at a temperature of −80 °C for further detection of serum levels of homocysteine, lipid profile, and renal function tests.

Homocysteine levels were measured by enzyme-linked immune sorbent assay (ELISA) using Human Homocysteine ELISA Kit catalogue # DEIA1724 following manufacturing instructions (Creative Diagnostics, Shirley, NY, USA). Lipid profile parameters included serum total cholesterol, triglyceride, HDL-cholesterol, and LDL-cholesterol. These parameters were assessed via enzymatic colorimetric method using the Hitachi 704 Analyzer (Roche Diagnostics, Indianapolis, IN, USA), while HDL-cholesterol levels were calculated by the Friedewald formula. Additionally, renal functions were determined using a fully automated clinical chemistry auto-analyzer Selectra proM, (ELITech Group, clinical chemistry automation systems, Finland).

### 2.3. Radiological Assessment

#### 2.3.1. Ultrasound Examination

##### Preparation

The children were not sedated before the ultrasound examination. The children who were too young to enable a thorough or comprehensive examination were eliminated from the research.

##### Equipment

Ultrasound examination was conducted using high-resolution B-mode ultrasound (Toshiba Medical Systems) equipped with a linear array transducer of 7 MHz.

##### Patient Position

Examination of the children was performed in the supine position while the neck was extended, and the head turned approximately 45 degrees to the left. The probe was in an antro-lateral position.

##### Scanning Technique

The right common carotid artery measurements were taken approximately one cm below the carotid bulb. Because it is a central artery that branches directly from the aorta, the common carotid artery (CCA) was chosen. The transducer was positioned to provide longitudinal visualization of the carotid artery. The distance between two nearly parallel echogenic lines separated by a hypoechoic zone shows the combined thickness of the intima and media (I–M complex) in longitudinal views of the CCA wall.

##### Equipment Settings

The focus depth was adjusted to between 30 and 40 mm. The gain settings were adjusted to enhance edge detection. The zoom function was used. At least 2–3 cardiac cycles (approximately 10 s) were recorded.

##### Echo-Tracking

Automatic measurements were obtained online or offline through an automatic software (Carotid Studio) for analysis of carotid intima-media thickness and carotid artery stiffness. The analysis is applicable to anybody above the age of 36 months. The system can handle video data that have been previously captured as well as video files collected in real-time from any ultrasound device. The carotid artery diameter was calculated using a contour-tracking approach applied to a B-mode ultrasound picture of a longitudinal portion of the artery as the distance between the far and near media-adventitia interfaces. A strong edge detection approach was also used to measure the IMT concurrently and automatically. Carotid-Studio was used to measure the CIMT and carotid artery diameter, which when paired with an estimate of arterial blood pressure, yields parameters of arterial elasticity and carotid artery stiffness.

### 2.4. Statistical Analysis

The collected data were coded, tabulated, and statistically analyzed using SPSS program (Statistical Package for Social Sciences) software version 25. Descriptive statistics were conducted for parametric (normally distributed) quantitative data by mean ± Standard Deviation (SD), for non-parametric quantitative data by median and interquartile range (IQR), and for qualitative data by frequency and percentage. Analyses were performed between the two groups for parametric quantitative data and between the four groups using a one-way ANOVA test. This was followed by post hoc analysis between the two groups, while the Kruskal–Wallis test was used for non-parametric quantitative data between the four groups, followed by the Mann–Whitney test between the two groups. Analyses were performed between the two groups for qualitative data using a chi-square test. Pearson’s and Spearman’s correlation coefficients were performed to detect the correlation between variables, followed by multiple stepwise linear regression analysis. The level of significance was taken at *p* value ≤ 0.05.

## 3. Results

This case-control study included 60 epileptic children on long-term ASMs therapy, and 25 children served as controls. Our cases were diagnosed with idiopathic epilepsy and received different ASMs for at least one year. They were all normotensive. Table 1 summarizes demographic data of the different studied groups. There was a significant difference between the studied groups regarding GASE scale (*p*-value = 0.02).

Serum levels of homocysteine and the lipid profile were significantly higher in children on both old generation monotherapy and polytherapy than that in children on new generation monotherapy and controls (*p*-value = 0.0001) with no significant difference between old generation monotherapy and polytherapy. There were no differences between all groups regarding blood urea and creatinine (Table 2).

Regarding carotid intima-media thickness (CIMT), it was significantly higher in children on old generation monotherapy and polytherapy than in children on new generation monotherapy and controls. On the other hand, there were no differences between all studied groups regarding carotid stiffness (Table 3) and (Figure 1).

Regarding serum levels of homocysteine, we analyzed its results and found that serum levels of homocysteine have positive correlation with LDL, TG, cholesterol, CIMT, carotid stiffness, and the duration of treatment (r = 0.472, 0.450, 0.474, 0.780, 0.602, and 0.712, respectively, with *p* < 0.0001). It was negatively correlated with HDL (r = −0.561, *p*-value < 0.0001) (Table 5).

Both CIMT and carotid artery stiffness were positively correlated with LDL, TG, cholesterol, and the duration of treatment. CIMT was negatively correlated with HDL (Table 6). 

With regards to multiple stepwise linear regression analysis, the variables which predict homocysteine level are HDL, TG, CIMT, and duration of treatment (R^2^ = 0.787) (Table 7).

## 4. Discussion

Many studies have shown that long term ASMs might cause certain adverse metabolic effects [8]. Various studies have found that these individuals had high serum levels of risk indicators, such as homocysteine [9,10]. Some studies found a progressive rise in total homocysteine levels with long-term ASMs usage and/or multiple ASMs therapy [11]. Although various research on the vascular adverse effects of ASMs has been conducted, the findings are contradictory [12]. 

Our study has shown that the mean serum level of homocysteine was significantly higher in children on monotherapy of old generation ASMs and also on polytherapy than in children on new generation monotherapy and controls, with no significant difference between old generation monotherapy and polytherapy. 

Old AEDs, as they are mainly enzyme inducers, may alter metabolic pathways related to vascular risk and may be associated with more adverse vascular effects. Enzyme inducer AEDs are associated with elevated serum levels of total cholesterol and low-density lipoprotein due to their effects on cholesterol synthesis enzymes [13] and C-reactive protein [14]. Sodium valproate is associated with weight gain and other metabolic effects that may increase the risk of cardiovascular events [14].

Verrotti et al., 2000, noticed a significant increase in homocysteine levels in pediatric patients treated with either carbamazepine (CBZ) or valproic acid (VPA) (old generation) when compared with controls [15]. Furthermore, Linnebank et al., 2011, found that older ASMs, such as carbamazepine, phenytoin, phenobarbital, and primidone, cause a deficit in folate, which is a key cofactor in the breakdown of homocysteine [16]. Huemer et al., 2005, found that higher homocysteine levels in patients receiving ASMs are predominantly associated with multidrug treatment [11].

On the other hand, Belcastro et al., 2010, indicated that recent ASMs, such as topiramate (TPM) and Oxcarbazepine (OXC), can produce excess homocysteine; however, lamotrigine (LTG) and levetiracetam (LEV) do not [17].

Disagreeing with our findings, Belcastro et al., 2010, and Emeksiz et al., 2013, found that there were no differences in Hcy levels among children receiving the VPA (old generation), OXC (new generation), and controls, and stated that new generation ASMs may cause an increase in serum homocysteine levels [17,18].

Sener et al., 2006, stated that plasma Hcy levels were not significantly different between patients on ASM polytherapy versus monotherapy [19]. 

In the current study, parameters of lipid profile serum levels were significantly lower in epileptic children on monotherapy of new generation ASMs and controls compared with that in children on monotherapy of old generation ASMs and polytherapy ASMs. There was no significant difference between serum lipid profile levels in children on monotherapy of old generation ASMs and polytherapy ASMs and no significant difference between monotherapy of new generation ASMs and controls. 

Our work is consistent with Franzoni et al., 2006, who demonstrated that OXC (new generation ASMs) does not alter the serum lipid profile in epileptic children when administrated [20]. El-Farahaty et al., 2015, stated that high serum levels of TC, LDL, and lower HDL levels were found in the valproic acid- and carbamazepine-treated groups compared to the controls [21]. Yamamoto et al., 2016, stated that patients using VPA showed a decrease in serum levels of HDL [12]. In his study, Calik et al., 2018, found that there is a significant increase in the lipid parameters in patients using multiple drug therapy for a long time [14].

Agreeing with our result, Safarpour et al., 2021, stated that low-density lipoprotein (LDL) values were greater in epileptic patients compared to controls, whereas high-density lipoprotein (HDL) values were reduced [22]. Conversely, Büyükgöl and Güneş, 2020, concluded that valproic acid and carbamazepine had no effect on lipoproteins that protect against coronary artery disease, such as HDL [23].

CIMT has been used in children to identify the onset of subclinical atherosclerosis. However, effort must be made to reduce CIMT measurement variance [24]. 

Measurement protocols of CIMT values widely differ among studies due to the lack of standardized protocols and inhomogeneous study populations, which make the comparison of results quite difficult [25]. Moreover, manual measurement of CIMT using B-mode ultrasound cannot identify such little variations in CIMT in pediatric age due to its limited sensitivity [26].

Using standard automated protocols for CIMT assessment, as we did in this study, reduces the variability associated with human error and enables study comparison [27]. 

In our study, we found a significant increase in CIMT and carotid stiffness in epileptic children who received old monotherapy of ASMs and polytherapy than in those who received new monotherapy and the controls, with no significant difference between groups of children on old monotherapy and polytherapy modalities of treatment. CIMT was positively correlated with TG, cholesterol, LDL, and duration of treatment. It was negatively correlated with HDL.

Both Karatoprak et al., 2020, and Kolekar et al., 2021, concluded that non-obese children with epilepsy who receive monotherapy of valproic acid or levetiracetam alone may be at higher risk of developing subclinical atherosclerosis despite a normal lipid profile [28,29]. This might be linked to atherosclerotic alterations, and these children will need to be closely monitored to avoid cardiovascular and cerebrovascular risks [29].

In 2015, Luo et al., found that the average CIMT of the epileptic patients treated with VPA was higher than that of healthy ones. Additionally, the average CIMT of patients with VPA administration with a duration of more than one year was higher than that of the patients with a VPA administration duration of less than one year [30]. Moreover, Yao-Chuang Chuang et al., 2012, indicated that prolonged monotherapy with the previous generation ASMs, e.g., CBZ, PHT, and VPA, resulted in considerably increased CIMT in patients with epilepsy, but no significant changes in CIMT were detected in patients receiving prolonged LTG monotherapy (new generation), and Oxcarbazepine therapy had no effect on carotid intima-media thickness when compared to the control group [31]. 

Calik et al., 2018, demonstrated that a significant increase in CIMT in patients on prolonged use of multiple drug treatments may indicate that these patients are at risk of developing vascular damage and other cardiovascular problems [14].

The role of ASMs in the pathogenesis of atherosclerosis and high CIMT is not completely understood [32]. Three major theories have been proposed: First, long-term ASM medication can cause dyslipidemia, which is marked by raised total cholesterol, low-density lipoprotein cholesterol (LDL-C), and lipoprotein A levels in the blood. LDL-C is known to have a role in the atherosclerotic process by increasing endothelial permeability, lipoprotein retention in the intima of blood arteries, inflammatory cell recruitment, and foam cell formation [32] and, hence, leads to an increase in CA-IMT [33,34]. Second, hyperhomocysteinemia might contribute to the development of atherosclerosis by increasing tumor necrosis factor expression, increasing oxidative stress and inducing a proinflammatory vascular state. Several investigations have indicated that enzyme-inducing ASMs can cause hyperhomocysteinemia by altering folate and vitamin B12 induction by liver enzymes. VPA may also be linked to higher blood homocysteine levels in epileptic patients (both adults and children), most likely through limiting folic acid absorption and directly interfering with folic acid coenzyme metabolism. LTG is a newer-generation ASM that does not induce enzymes or raise blood homocysteine levels, suggesting that it may have a small role in the atherosclerotic process [35,36,37]. 

Third, the inflammatory marker c-reactive protein (CRP) was shown to be higher in epileptic patients on ASM treatment. CRP has been shown in previous research to induce inflammation and atherosclerosis via influencing monocytes and endothelial cells, as well as boosting the activity of plasminogen activator inhibitor 1 [33].

In contrast to our study, Lai et al., 2017, noticed that no significant difference in CIMT was detected in children using ASMs [32]. Additionally, Keenan et al., 2014, also stated that there were no differences in mean CIMT between controls and children using ASMs [38]. Differences in age groups, duration of disease, and treatment collected with individual variation in response to treatment may explain this dissimilarity in results.

Arterial stiffness is as important as CIMT to be evaluated. Several terms were used synonymously with arterial stiffness, such as compliance, distensibility, and elasticity. There are several calculations that explain somewhat different arterial stiffness parameters. The artery wall has features that make it behave like both an elastic solid and a viscous liquid [39].

Unexpectedly, we did not find a significant difference in carotid stiffness among the different groups in our study. This means that stiff arteries are not always related to increased intima-media thickness. Weberruß et al., 2015, reported that higher arterial compliance and lower stiffness were associated with greater intima-media thickness. This can be attributed to a functional arterial adaptation in the pediatric age group, where an increased intima-media thickness is not related to arterial stiffness [40].

These findings support the theory that thickening at lower levels of CIMT reflects an equilibrium condition in which the effects of pressure and flow on the arteries are balanced. It may still be used as a graded marker for cardiovascular risk after it reaches a particular threshold [41].

To our knowledge, the current study is one of the very few studies conducted on epileptic children taking ASMs, either old or new drugs or a combination of both. Most of the previous studies regarding this issue were about adults with epilepsy. Therefore, this aspect gives our study a powerful point of view. A major worry about the prescription of ASMs is whether or not these medications lead to the development of atherosclerosis in epileptic children, and about their effects on CIMT, hence, provoking the incidence of cardiovascular complications. As per our findings, the risk of cardiovascular diseases is significantly augmented in epileptic children receiving enzyme-inducing ASMs (old generations) than non-enzyme inducing ones (new generations). Accordingly, this study will add in quantifying the risk of incidence of cardiovascular complications in epileptic children under enzyme-inducing ASMs; therefore, physicians should be very careful when prescribing these drugs, avoiding them as much as possible, and if their usage is a must, monitoring the risk indicators of cardiovascular diseases should be routine.

However, there are some limitations in our study, which should be considered in future studies. The sample of studied children was limited to one center. A multicenter study containing Egyptian children is recommended. Furthermore, it was suggested that the heterogeneity between the studied population may play a role in the effect of ASMs and tailored therapy should be applied. Thus, more studies should be conducted on different population groups. Secondly, future research should be performed to compare epileptic children who are on ASMs against those who are not, which is missed in the current study. Moreover, the length of ASM exposure varied between children in this study, which might have had implications on our findings and should be limited as much as possible in any future research directions. Moreover, prospective studies involving the follow up of epileptic children since their diagnosis, starting before the initiation of ASM treatment and then continuing after therapy, should be conducted. Finally, evaluating the protein level plus the genomic expression of more different vascular endothelial dysfunction and cardiac risk biomarkers along with radiological assessment could be helpful.

## 5. Conclusions

Our findings suggest that pediatric neurologists who prescribe ASMs should be cognizant of possible negative side effects, particularly in children who are at high risk of vascular problems and the use of old/combined drug therapy. Children with additional comorbidities, such as obesity, diabetes mellitus, dyslipidemia, and children whose parents have cardiovascular risk factors, might benefit from our results.

## Figures and Tables

**Figure 1 children-09-01499-f001:**
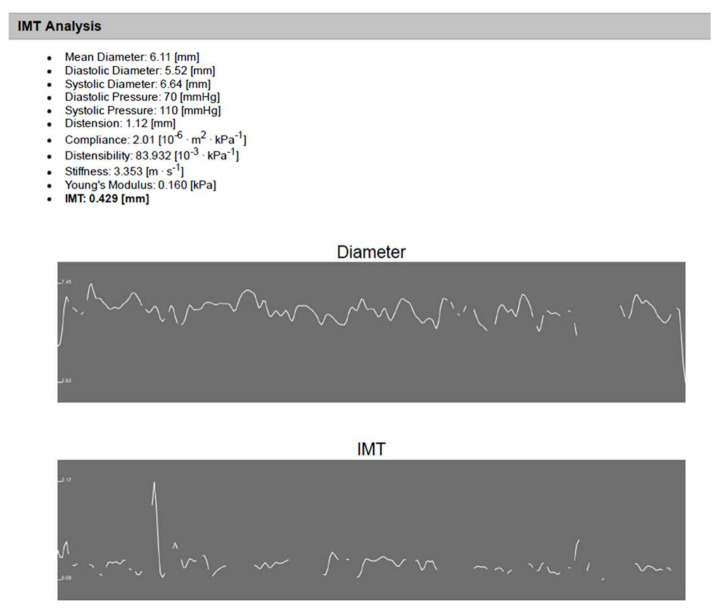
An illustrative example of IMT analysis using (Carotid Studio) software for an 11 year old child on old ASMs for 7 years. The analysis shows automatic measurement of CCA diameter and IMT as well as automatic calculated arterial stiffness.

**Table 1 children-09-01499-t001:** Demographic data of children within studied groups (*n* = 85).

	Group A *N* = 20	Group B *N* = 20	Group C *N* = 20	Group D *N* = 25	*p* Value
Age (year)					
Median	7	5.8	6.5	6	
Interquartile range	6–9	5.5–12	4.6–8	4.5–10	0.72
Sex					
Male	10 (50%)	9 (45%)	12 (60%)	13 (52%)	
Female	10 (50%)	11 (55%)	8 (40%)	12 (48%)	0.817
BMI					
Normal BMI	20 (100%)	20 (100%)	20 (100%)	25 (100%)	ــــــــــــ
Blood pressure (mmHg)					
Normal	20 (100%)	20 (100%)	20 (100%)	25 (100%)	ــــــــــــ
Duration of treatment (year)					
Median	2.5	2	3		
Interquartile range	2–3.8	1–3	2–4	ـــــــــــــــ	0.083
GASE scale					
A little severe	5 (25%)	12 (60%)	2 (10%)		
Somewhat severe	7 (35%)	6 (30%)	8 (40%)		
Moderate severe	6 (30%)	2 (10%)	8 (40%)	ـــــــــــــــ	
Quite severe	2 (10%)	0 (0%)	2 (10%)		0.020

Group A: epileptic children received a single old antiseizure medication. Group B: epileptic children received a single new antiseizure medication. Group C: epileptic children received two or more ASMs whatever its generation. Group D: control group. N: number; GASE: Global Assessment of Severity of Epilepsy Scale; BMI: Body Mass Index.

**Table 2 children-09-01499-t002:** Comparison between studied groups with regards to laboratory data.

	Group A *N* = 20	Group B *N* = 20	Group C *N* = 20	Group D *N* = 25	*p* Value	
Homocysteine (µmol/L)					A versus B A versus C A versus D B versus D C versus D B versus C	0.0001 0.9 0.0001 0.8 0.0001 0.0001
Median	16	2.3	17	1.7		
Interquartile range	15–18.7	0.8–5.8	11.5–20.7	1.2–4		
Blood Urea (mg/dL)					A versus B A versus C A versus D B versus D C versus D B versus C	0.3 0.1 0.7 0.4 0.08 0.3
Mean ± SD	25.7 ± 1.8	25.8 ± 1.03	26.1 ± 0.8	25.6 ± 1.07		
Creatinine (mg/dL)					A versus B A versus C A versus D B versus D C versus D B versus C	0.4 0.4 0.5 0.7 0.8 0.9
Mean ± SD	0.5 ± 0.1	0.5 ± 0.08	0.5 ± 0.07	0.5 ± 0.08		
Cholesterol (mg/dL)					A versus B A versus C A versus D B versus D C versus D B versus C	0.0001 0.8 0.0001 0.6 0.0001 0.0001
Median	195	115	200	125		
Interquartile range	180–271.2	100–130	161.2–265	92.5–140		
LDL (mg/dL)					A versus B A versus C A versus D B versus D C versus D B versus C	0.0001 0.9 0.0001 0.6 0.0001 0.0001
Median	132	47	144	53		
Interquartile range	114–204.2	33.2–62.7	91.5–174.2	28–77		
HDL (mg/dL)					A versus B A versus C A versus D B versus D C versus D B versus C	0.0001 0.3 0.0001 0.04 0.0001 0.0001
Mean ± SD	33 ± 6.5	49.7 ± 2.6	30.8 ± 5.4	49.4 ± 10.5		
TG (mg/dL)					A versus B A versus C A versus D B versus D C versus D B versus C	0.0001 0.8 0.0001 0.04 0.0001 0.0001
Median	175	97.5	175	90		
Interquartile range	131.2–190	90–110	110–193.7	47.5–110		

LDL: low-density lipoprotein; HDL: high-density lipoprotein; TG: Triglyceride.

**Table 3 children-09-01499-t003:** Radiological results of the studied groups.

	Group A *N* = 20	Group B *N* = 20	Group C *N* = 20	Group D *N* = 25	*p* Value	
CIMT (mm)					A versus B A versus C A versus D B versus D C versus D B versus C	0.0001 0.3 0.0001 0.4 0.04 0.01
Mean ± SD	0.46 ± 0.03	0.36 ± 0.08	0.43 ± 0.1	0.38 ± 0.08		
Carotid stiffness (cm/s)					A versus B A versus C A versus D B versus D C versus D B versus C	0.06 0.2 0.1 0.3 0.6 0.4
Mean ± SD	4.5 ± 0.8	3.9 ± 0.8	4 ± 1.1	4.2 ± 0.8		

CIMT: Carotid intima-media thickness. GASE scale was positively correlated with homocysteine, CIMT, carotid stiffness, LDL, TG, cholesterol serum levels, and the duration of treatment (r = 0.699, 0.67, 0.54, 0.36, 0.39, 0.36, and 0.59, respectively, with *p* = 0.001, 0.0001, 0.0001, 0.004, 0.002, 0.004, and 0.001, respectively). On the other hand, it was negatively correlated with HDL (r = −0.31, *p*-value = 0.01) (Table 4).

**Table 4 children-09-01499-t004:** Correlation between GASE scale and different parameters in the studied patients (*n* = 60).

GASE Scale		
Parameters	R	*p* Value
Homocysteine	0.699	<0.001
CIMT	0.67	0.0001
Carotid stiffness	0.54	0.0001
LDL	0.36	0.004
HDL	−0.31	0.01
TG	0.39	0.002
Cholesterol	0.36	0.004
Duration of treatment	0.59	<0.001

r = 0.75–1 (strong correlation), r = 0.5–0.74 (moderate correlation), r = 0.25–0.49 (fair correlation), r = 0.1–0.24 (weak correlation).

**Table 5 children-09-01499-t005:** Correlation between homocysteine and different parameters in the studied patients (*n* = 60).

Homocysteine		
Parameters	R	*p* Value
LDL	0.472	<0.001
HDL	−0.561	<0.001
TG	0.450	<0.001
Cholesterol	0.474	<0.001
CIMT	0.780	<0.001
Carotid stiffness	0.602	<0.001
Duration of treatment	0.712	<0.001

**Table 6 children-09-01499-t006:** Correlation between carotid thickness and stiffness with different studied parameters (*n* = 60).

CIMT			Carotid Stiffness	
Parameters	R	*p* Value	R	*p* Value
LDL	0.39	0.002	0.29	0.02
HDL	−0.39	0.002	−0.19	0.1
TG	0.31	0.01	0.33	0.01
Cholesterol	0.41	0.001	0.33	0.009
Duration of treatment	0.64	0.0001	0.40	0.001
Carotid stiffness	0.76	0.0001		

**Table 7 children-09-01499-t007:** Multiple stepwise linear regression analysis of parameters predicting homocysteine levels revealed the best prediction model.

	Unstandardized Coefficients		*p* Value	Adjusted R^2^	SEE
	(Constant)	B			
CIMT (mm)	−3.519	44.397	<0.001	0.787	5.3
HDL (mg/dL)		−0.347	<0.001		
Duration of treatment (years)		2.123	<0.001		
TG (mg/dL)		0.036	0.013		

## Data Availability

The datasets generated and/or analyzed during this study are not publicly available However, the datasets can be shared by the corresponding author upon request.

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
