# Peer review of "Influence of Different Antiseizure Medications on Vascular Risk Factors in Children with Epilepsy"

_children, 2022, doi:10.3390/children9101499_

Round 1
Reviewer 1 Report
The topic of this manuscript is interesting and useful result for children with epilepsy the appropriate choice of antiseizure medicine.
But there were several problems as follows:
Introduction:
1. There were not enough previous publications to support the correlation of anti-seizure medicine (ASM) and the vascular risk factors. Any publications supported the old generations had the high risk than new generation ASM? What’s possible mechanism
2. There were less evidence of previous data focus on children.
Materials and Methods
1. Patients’ age form 2-15 years, why did you choice young baby for the vascular survey? Is the method of carotid duplex exam difference of baby compared with children?
2. In Group 3, is polytherapy included old and new generation medicines?
Results:
1. The serum levels of homocysteine and lipid profile were significant difference in each groups, that’s too obvious. Is there any possibility of selection bias? Is there any other factors contributed the results beside the ASM factor.
2. The CIMT of group 1 is higher than other groups, the measurement is just taken one point of right CCA, is that represented the whole carotid artery condition ?
Discussion:
1. About the level of homocysteine and lipoprotein (LDL and HDL), authors use many different articles to agree or against the results of this article, but there were less evidences to mention why the difference of this article between old and new generation ASMs so significant. Is there any co-factors that contributed this results.
2. In page 9, line 259, misspelling “groupd”
3. About the CIMT of patients with each different ASMs, is any strong evidences of children that can support author’s results? Is any different findings , mechanisms, or outcomes between children and adults ?
4. I agreed with the limitations of this articles included heterogeneity in the study population and length of ASM exposure varied greatly. But I suggested that should add some explanations in the result or discussion.
Reviewer 2 Report
Comment 1: The old abbreviation (AED) should be replaced with ASM, which means “antiseizure medication”.
Comment 2: The type of ASMs (e.g. levetiracetam, carbamazepine, valproate…) should be mentioned for each epileptic group.
Comment 3: Sample size calculation/power calculation should be included.
Comment 4: How many months/years after the first diagnosis of epilepsy did you collect the blood sample to measure the total homocysteine levels? Are the patients with newly diagnosed epilepsy or not?
Comment 5: Did you check whether the total homocysteine levels were significant different before vs after the treatment with ASMs?
For instance, a number of neuropeptides have been implicated with the pathogenesis of epilepsy. Several studies demonstrated that ghrelin may play a significant role in epilepsy. Particularly, ghrelin and the cognate peptide des-acyl ghrelin (DAG) were significantly higher in the plasma of children positively responding to ASMs (responders) [1], raising the question of whether these hormones could be increased by the administration of ASMs, or alternatively, the children committed to be responders to ASMs could be characterized by pre-existing higher ghrelin plasma levels [2]. Then, it was suggested that total ghrelin plasma levels and the ghrelin-to-DAG ratio were unchanged by ASMs, but they were determined by demographic and clinical features such as the type of epilepsy, age, head circumference, and BMI [3]. Could you please discuss this possibility?
1) Marchiò, M.; Roli, L.; Giordano, C.; Caramaschi, E.; Guerra, A.; Trenti, T.; Biagini, G. High Plasma Levels of Ghrelin and Des-Acyl Ghrelin in Responders to Antiepileptic Drugs. Neurology 2018, 91, e62–e66.
2) Tollefson, T.J.; Berg, M.J. Comment: Ghrelin and Des-Acyl Ghrelin: Do They Predict Success of AED Treatment? Neurology 2018, 91, 29.
3) Costa, A.-M.; Lo Barco, T.; Spezia, E.; Conti, V.; Roli, L.; Marini, L.; Minghetti, S.; Caramaschi, E.; Pietrangelo, L.; Pecoraro, L.; et al. Prospective Evaluation of Ghrelin and Des-Acyl Ghrelin Plasma Levels in Children with Newly Diagnosed Epilepsy: Evidence for Reduced Ghrelin-to-Des-Acyl Ghrelin Ratio in Generalized Epilepsies. J. Pers. Med. 2022, 12, 527. https://doi.org/10.3390/jpm12040527
Reviewer 3 Report
This study investigates the vascular risk factors, including serum homocysteine level in pediatric epilepsy patients who received a long-term anti-epileptic drugs. The results show that patients with old generation monotherapy or polytherapy had significantly higher homocysteine level and lipid profiles.
This is a relatively simple study that includes food number of patients in each subgroup. It would be of interest to discuss how the findings may have a possible impact on the treatment of epilepsy. Also it may be informative to see what kind of future studies are useful to address the limitations, other than increasing the number of patients.
Round 2
Reviewer 2 Report
The authors addressed all my concerns. Please, just check again AED!